# Paralogues of Mmp11 and Timp4 Interact during the Development of the Myotendinous Junction in the Zebrafish Embryo

**DOI:** 10.3390/jdb7040022

**Published:** 2019-12-03

**Authors:** Emma F. Matchett, Shuaijin Wang, Bryan D. Crawford

**Affiliations:** Biology Department, University of New Brunswick, Fredericton, NB E3B 5A3, Canada; emma.f.matchett@gmail.com (E.F.M.); wangjenny64@gmail.com (S.W.)

**Keywords:** stromelysin-3, Mmp11, Timp4, furin, myotendinous junction, extracellular matrix remodeling, zebrafish, post-translational regulation, EMMA assay

## Abstract

The extracellular matrix (ECM) of the myotendinous junction (MTJ) undergoes dramatic physical and biochemical remodeling during the first 48 h of development in zebrafish, transforming from a rectangular fibronectin-dominated somite boundary to a chevron-shaped laminin-dominated MTJ. Matrix metalloproteinase 11 (Mmp11, a.k.a. Stromelysin-3) is both necessary and sufficient for the removal of fibronectin at the MTJ, but whether this protease acts directly on fibronectin and how its activity is regulated remain unknown. Using immunofluorescence, we show that both paralogues of Mmp11 accumulate at the MTJ during this time period, but with Mmp11a present early and later replaced by Mmp11b. Moreover, Mmp11a also accumulates intracellularly, associated with the Z-discs of sarcomeres within skeletal muscle cells. Using the epitope-mediated MMP activation (EMMA) assay, we show that despite having a weaker paired basic amino acid motif in its propeptide than Mmp11b, Mmp11a is activated by furin, but may also be activated by other mechanisms intracellularly. One or both paralogues of tissue inhibitors of metalloproteinase-4 (Timp4) are also present at the MTJ throughout this process, and yeast two-hybrid assays reveal distinct and specific interactions between various domains of these proteins. We propose a model in which Mmp11a activity is modulated (but not inhibited) by Timp4 during early MTJ remodeling, followed by a phase in which Mmp11b activity is both inhibited and spatially constrained by Timp4 in order to maintain the structural integrity of the mature MTJ.

## 1. Introduction

The development of skeletal muscle is highly conserved in vertebrates, with the bulk of the musculature arising from the somitic mesoderm, which differentiates into segmented myotomes early in embryonic development [1]. Somites are patterned anterior-posteriorly by a clock-and-wave-front mechanism [2,3] and become delineated by boundaries that form at the interface between cells with high and low cadherin-mediated cell–cell adhesion, into which a fibronectin-based extracellular matrix (ECM) is deposited [4]. Like many provisional fibronectin-rich matrices, the ECM between developing somites is replaced by a more complex matrix consisting of laminin-rich basement membranes flanking a core rich in fibrillar collagens as the somites mature into myotomes, and as the rectangular somite boundaries mature into chevron-shaped myotendinous junctions (MTJs) [1,5,6,7]. This biochemical and structural remodeling of the ECM is essential for the normal development of skeletal muscle and is an ideal context in which to study developmentally regulated matrix remodeling in vivo.

Matrix metalloproteinases (MMPs) are well known for their roles in matrix remodeling [8,9,10,11]. In zebrafish, Mmp11 (Stromelysin-3) is both necessary and sufficient for the removal of fibronectin at the maturing MTJ, and this process is activated by the polymerization of laminin at the MTJ [12]. Stromelysin-3 was first described in human breast carcinomas [13] and its expression is an important prognostic indicator in many cancers [14]. Several studies have implicated Stromelysin-3 as an important effector of matrix remodeling during amphibian metamorphosis [15,16,17,18,19], but frustratingly little else is known about its roles in normal physiology and development. Stromelysin-3-deficient mice are fertile and normal in both behavior and appearance [20]. However, redundancy in the mammalian Stromelysin gene family may be obscuring essential developmental roles in these knockouts. While the zebrafish has duplicate paralogues of genes encoding Stromelysin-3 (designated *mmp11a* and *mmp11b*), it lacks orthologues of other members of the Stromelysin gene family [21], potentially making it a better model organism in which to study the functions of Stromelysin-3 in vivo. Furthermore, its rapid external development as a transparent embryo, amenability to molecular techniques (including specific assays for the detection of MMP activation and activity [22,23,24,25]), and various other characteristics has made the zebrafish a favorite model organism for the study of muscle development and disease, as well as MMP activity and matrix remodeling [1,7,9,21,26,27,28,29,30].

Underscoring their importance in development and physiology, the regulation of MMP activity is extraordinarily complex and nuanced, comprising mechanisms acting at transcriptional, post-transcriptional, translational, and several post-translational levels [11,31,32]. All MMPs are translated as inactive proMMPs (zymogens) with auto-inhibitory N-terminal pro-peptides containing an unpaired cysteine residue that folds back into the active site and co-ordinates with the catalytic zinc ion, rendering the protease inactive [11,33]. The proteolytic removal of this propeptide, or the chemical modification of the thiol side chain in the ‘cysteine switch’ renders the protease active [11,34,35,36,37]. Most MMPs are activated by other extracellular/cell-surface proteases, including other MMPs, but Stromelysin-3 has a conserved paired basic amino acid cleavage site that is thought to be recognized and cleaved by furin during its transit through the trans-Golgi network, and is therefore expected to be secreted in its active form [38,39]. Once activated, interaction with tissue inhibitors of MMPs (TIMPs) or other endogenous inhibitors maintains a check on unregulated proteolytic activity [11,40,41,42]. Early in vitro studies of Stromelysin-3 suggested it was bound and inhibited by TIMPs 1 and 2 [38]. However, our understanding of the interactions between TIMPs and MMPs in vivo remains extremely limited [41] and nothing is known about the interaction of Stromelysin-3 with TIMPs in vivo.

Furthermore, despite their widely assumed importance in degrading extracellular matrix proteins, the substrates MMPs degrade in vivo also remain poorly characterized [43,44]. Recent work has revealed a plethora of non-matrix and even intracellular substrates, suggesting these proteases may be functioning in many unexpected ways [45,46,47,48,49]. Stromelysin-3 exhibits little proteolytic activity against matrix proteins in vitro [39,50,51,52], but it can degrade protease inhibitors and the insulin-like growth factor binding protein in biochemical assays [53]. The only substrate Stromelysin-3 is known to act on in vivo is the 67 kD cancer-associated non-integrin laminin receptor, which it degrades during tadpole metamorphosis [18,39,54,55,56], suggesting that like other MMPs, it may play important roles in signal transduction and modulating the activity of other players in the cellular microenvironment. Furthermore, there is a growing body of evidence that many MMPs (including Stromelysin-3) may accumulate and function intracellularly [47,48,49,57,58,59,60,61,62], suggesting that matrix remodeling is only one of many important roles played by these proteases.

As with the duplicated paralogues of Stromelysin-3, the zebrafish genome includes duplicate paralogues of genes encoding tissue inhibitors of metalloproteinase-4 (designated *timp4a* and *timp4b*) [21]. Orthologues of *timp4* are present in all vertebrate genomes that have been examined. However, little is known about this endogenous MMP inhibitor. Here, we show that the protein products of both zebrafish *mmp11* paralogues localize to the MTJ during myotome maturation, but in temporally reciprocal patterns. In addition, we note that like Mmp2, Mmp11a accumulates intracellularly within the sarcomeres of skeletal muscle. However, in contrast to Mmp2 which accumulates in the M-lines of sarcomeres [49], Mmp11a localizes to the Z-discs. Furthermore, despite having a weaker furin recognition motif in its propeptide than Mmp11b, Mmp11a is activated by furin as it transits the secretory pathway. However, we also find evidence of furin-independent activation intracellularly within the nuclei of some cells. Timp4 is also present in the MTJ throughout the developmental period examined, and we find that domains of both paralogues of Timp4 interact with domains of both Mmp11 paralogues, but with distinctly different specificities. Considering these data alongside sequence analysis and structural homology models, we propose that the duplicated paralogues of Mmp11 have diverged and play distinct roles in both the developmental remodeling and subsequent maintenance of the MTJ and, furthermore, that Timp4 paralogues have similarly diverged to independently modulate Mmp11 activity at the maturing MTJ.

## 2. Materials and Methods

### 2.1. Animal Care and Spawning

Zebrafish were maintained at a 14 h light/10 h dark cycle at 28 °C on a flow-through rack system and fed a standard zebrafish diet of GEMMA 500 twice daily, and brine shrimp once a day. Tübingen (wild type) adults were placed in breeding tanks tilted to mimic shallow spawning environment, and dividers were placed to separate males and females for the purpose of controlling spawning time and thereby synchronizing embryo development. Embryos were collected 30 min after the beginning of the light cycle, maintained in embryo rearing medium (ERM) (1.37 mM NaCl, 54 µM KCl, 2.5 µM Na_2_HPO_4_, 4.4 µM KH_2_PO_4_, 13 µM CaCl_2_, 10 µM MgSO_4_, 42 µM NaHCO_3_, pH adjusted to 7.2 with NaOH) with 0.0001% methylene blue to inhibit fungal growth and staged according to [63]. Chorions were removed manually using fine forceps. All work with zebrafish was done with the approval and under the supervision of the University of New Brunswick’s Animal Care Committee (UNB Animal Care Protocols 15016, 16018 and 17016) in accordance with Canadian Council on Animal Care guidelines.

### 2.2. Immunostaining and Microscopy

Embryos of specified stages were incubated overnight in Dent’s fixative (80% methanol, 20% dimethyl sulfoxide (DMSO)) for immunostaining using antibodies against Mmp11, Timp4, Laminin and α-actinin, or 4% formaldehyde in ERM for immunostaining with antibodies against hemagglutinin (HA) and GFP epitopes in epitope-mediated MMP activation (EMMA) assays. Embryos were washed 3× with phosphate buffered saline (PBS) containing 0.1% triton X-100 (PBSTx) and incubated overnight in blocking buffer (PBSTx + 5% bovine serum albumin (BSA)). Embryos were incubated in primary antibodies (mouse anti-α-actinin (catalogue #A7811; Sigma, Oakville, ON Canada), rabbit anti-laminin (catalogue #PA1-16730; ThemoFisher Scientific, Waltham, Massachusetts, USA) mouse anti-GFP (catalogue #11814460001, Roche, Basel, Switzerland), rat anti-HA (Roche catalogue #1186742300), rabbit anti-zebrafish-Mmp11a (catalogue #55688; AnaSpec, Freemont, CA, USA), rabbit anti-zebrafish-Mmp11b (AnaSpec catalogue #55690), and rabbit anti-Zebrafish-Timp4 (AnaSpec catalogue #55501)) diluted 1:1000 in blocking buffer overnight. Embryos were washed 3× in PBSTx and incubated overnight with secondary antibodies (goat anti-Mouse Alexa488, donkey anti-Rat Alexa594 and goat anti-Rabbit Alexa-488 (Invitrogen, Carlsbad CA, USA)), diluted 1:1000 in blocking buffer. Embryos were washed 3× and mounted in PBSTx and visualized using a Lecia SP2 fitted with a 20× 0.7NA water immersion and 63× 1.4NA oil immersion objectives. All washes were performed for 15 min each, and overnight incubations were at 4 °C.

### 2.3. Construction of EMMA–Mmp11a Expression Vector

The EMMA–Mmp11a expression vector was generated using Gateway^TM^ (Version E) (Invitrogen) cloning by amplifying the coding sequence from *mmp11a* cDNA (Open Biosystems; EDR1052-99562639) using forward (GGGGACAAGTTTGTACAAAAAAGCAGGCTTACACAAAGAGTCCCCGGGTTT) and reverse (GGGACCACTTTGTACAAGAAAGCTGGGTAGAAGAAGTCAGCACCGAT) primers designed to generate an amplicon that excluded the endogenous *mmp11a* secretory signal sequence and the stop codon. Entry clones were generated by recombining 50 ng purified PCR product with pDONR^TM^221 using BP Clonase overnight at RT. Positive colonies were verified by colony PCR using *mmp11a* specific primers (forward: TTTACAAGGTACTGGCATGG; reverse: TTGAGAGGATAGGAGAAGCTG). The final expression vector was generated by recombining 2 µL verified *mmp11a* entry vector with 2 µL empty EMMA Destination vector [25] using LR Clonase overnight at RT. All plasmids were verified by sequencing (Robarts) before use.

### 2.4. Microinjection and Heat Shock

Embryos were collected immediately post fertilization at the one-cell stage and microinjected with 3–5 nl of plasmid diluted in Danieau buffer (50 mM NaCl, 0.7 mM KCl, 0.4 mM MgSO_4_, 0.6 mM Ca(NO_3_)_2_, 5 mM 2-[4-(2-hydroxyethyl)piperazin-1-yl]ethanesulfonic acid (HEPES) (pH 7.6)) to a final concentration of 150 ng/mL (i.e., individual embryos would be injected with 350–750 fg of plasmid DNA) using a pressure-based microinjection apparatus (ASI), and incubated at 28 °C in ERM for 6 h. Normally developing embryos were transferred to fresh ERM and incubated at 28 °C until 24 hours post fertilization (hpf), at which time the expression of the injected construct was induced by transferring embryos to a 2 mL microcentrifuge tube at 28.5 °C, increasing the temperature to 37 °C over 5–7 min, maintaining at 37 °C for 30 min, and then returning to 28.5 °C over 10 min. At 3 to 6 h post heat shock (hpHS), which corresponds to ~28 to 31 hpf, GFP fluorescence was visualized using a Leica M205FA stereo epifluorescent microscope, and embryos expressing the construct were fixed in 4% formaldehyde or homogenized for immunoblotting.

### 2.5. Embryo Homogenization, SDS-PAGE, and Immunoblotting

Embryos were de-yolked on ice with a p200 pipette in 10% EDTA and 0.003% Phenylmethanesulfonyl fluoride (PMSF) in ERM, and homogenized in a 1.5 mL microcentrifuge tube with 10 µL 2× SDS-PAGE buffer (125 mM Tris pH 6.8, 20% glycerol, 4% SDS, 0.05% bromophenol blue, 10% β-mercaptoethenol) per embryo, heated to 95 °C for 4 min, and insoluble debris removed by centrifugation. Identically loaded samples were resolved on duplicate 10% acrylamide gels at 120 V for 1.5 h. One gel was transferred to PVDF membrane at 90 V for 1 h while the other was silver stained and imaged immediately. Non-specific binding to the membrane was blocked using 3% skim milk powder dissolved in PBS with 0.1% Tween-20 (PBSTw) for 1 h at RT, and bound proteins were detected using mouse anti-GFP (Roche) diluted 1:5000 in blocking buffer overnight at 4 °C, followed by washing thrice in PBSTw, incubating in horseradish peroxidase (HRP)-conjugated goat-anti-mouse diluted 1:10,000 in blocking buffer for 1 h at RT, washing thrice, and finally imaging using Luminata^TM^ Crescendo ECL (catalogue # WBLUR0020; Millipore, Etobicoke, Ontario Canada) detection reagent.

### 2.6. Furin Inhibition Treatment

Embryos were raised for 24 h after injection with the EMMA–Mmp11a construct and heat shocked as described above, but with the addition of 50 nM furin inhibitor I (Decanoyl-RVKR-CMK, catalogue# 3501/1; R&D Systems, Minneapolis, MD, USA), or vehicle (0.1% DMSO) to the ERM in which they were incubated.

### 2.7. Yeast Two-Hybrid Assay

Yeast two-hybrid assay of specific interactions between domains of Mmp11 and Timp4 paralogues was conducted according to the Yeast Two-Hybrid ProQuest^TM^ System (Invitrogen) protocol. Briefly, Bait (pDEST^TM^32) and Prey (pDEST^TM^22) plasmids were recombined with PCR amplicons generated using primers containing Att sites, which comprised the open reading frames of both *mmp11s* and both *timp4s* paralogs. These were further refined by deletion to encode single domains of each, i.e., catalytic or hemopexin domain for each *mmp11* paralogue or C-terminal or N-terminal domains of each *timp4* paralogue. Platinum^TM^ DNA Polymerase (Invitrogen) or Phusion^TM^ DNA Polymerase (NEB) was used to ensure high fidelity and confirmed by sequencing (Robarts).

Yeast MaV203 cells were transformed with engineered Bait and Prey plasmids, and successful transformations were confirmed by colony growth on selective media lacking leucine and tryptophan (SC-Leu-Trp). Transformants were plated on various section media, SC-Leu-Trp-His, SC-Leu-Trp-Ura, SC-Leu-Trp-Ura-His and SC-Leu-Trp-His+3AT (3 mM or 12 mM, comprising low and high stringency, respectively), to assay for interactions between the fusion proteins encoded by the Bait and Prey plasmids.

### 2.8. In silico Protein Modeling

Protein models were generated using SWISS-MODEL (https://swissmodel.expasy.org/), the inferred amino acid sequence of zebrafish Mmp11a, Mmp11b, Timp4a and Timp4b and the following known structures were provided as templates: the crystal structure of the mouse MMP-11 catalytic domain (1HV5), the crystal structure of the human MMP-1 hemopexin domain (2CLT), the crystal structure of human proMMP-12 (3BA0), and the crystal structure of human MMP-13 (4FU4). For the TIMP models, two different structures representing human TIMP-2 complexed to MMP-10 (stromeylisin-2) and proMMP-2 (4ILW and 1GXD, respectively) were used as templates. Software parameters were left at default settings for the prediction of eukaryotic protein structures, and the resulting PDB files were rendered using Chimera (version 1.12, http://www.rbvi.ucsf.edu/chimera/) [64].

## 3. Results

### 3.1. Zebrafish Mmp11 and Timp4 Sequences Exhbit Both Conserved and Divergent Features

Both zebrafish *mmp11* paralogues encode proteins with easily recognizable secretory signals, auto-inhibitory propeptides (with conserved cysteine switches), furin recognition motifs, catalytic domains (with conserved zinc-binding sites), hinge regions, and carboxyl hemopexin-like domains (Figure 1A), and structural homology models of both proteins are strikingly similar (Figure 1B). However, there are some notable differences that may be functionally significant. The furin recognition motif is not completely conserved (RQKR vs. RRKR in Mmp11a and Mmp11b, respectively), suggesting that furin-mediated activation of Mmp11a may be less efficient. Perhaps more significantly, differences in the specificity and S-loops [65] suggest that the specificity and/or affinity of these proteases for substrates is likely not identical. Finally, the disordered hinge or ‘linker’ region connecting the catalytic and hemopexin-like domains is sixteen residues (~76%) longer in Mmp11b. MMPs are thought to work their way along fibrillar substrates inchworm-fashion by binding alternately with the catalytic and then hemopexin-like domains [66]. So, differences in linker lengths may substantially alter the spacing of cut sites even if both of these proteases hydrolyze some of the same substrates.

The paralogous zebrafish *timp4* genes also encode proteins with easily recognizable canonical domains (Figure 1C), but there is more obvious divergence, particularly in the carboxyl domain, which is significantly smaller in Timp4a (Figure 1D). Timp4a also includes a highly charged seven-residue insertion in the N-terminal domain that is likely to affect protein-protein interactions.

### 3.2. The Propeptide of Mmp11a is Removed by Furin Proprotein Convertase

Since its initial characterization, researchers have assumed that Stromelysin-3 is activated by ER/Golgi-resident furin proprotein convertases during its transit through the secretory pathway due to the conserved paired basic amino acid furin recognition sequence present at the carboxyl end of the propeptide [38,39]. However, this has never been verified in vivo, and the differences in the furin recognition sequences present in the zebrafish Mmp11 paralogues led us to wonder if these proMMPs are both effectively activated by furin, and furthermore, whether proteolytic removal of the propeptide by furin is the sole activating mechanism for Mmp11 proteins in zebrafish. To investigate these questions, we used transient epitope-mediated MMP activation (EMMA) assays [25] to assess the effect of the strong furin inhibitor Decanoyl-RVKR-CMK on the proteolytic activation of the Mmp11 paralogue with the weaker furin recognition motif (i.e., RQKR in Mmp11a).

The principle of the EMMA assay is that the MMP in question is expressed as a doubly-epitope tagged fusion protein bearing the hemagglutinin (HA) tag near the N-terminus (but carboxyl to the secretory signal such that it remains present on the propeptide after translation), and a GFP moiety at the carboxyl terminus, allowing proteolytic removal of the propeptide to be detected in situ by loss of co-localization of the HA and GFP epitope tags, and in vitro by immunoblotting (Figure 2).

We generated a construct encoding EMMA–Mmp11a, expressed it transiently in embryos under control of the heat shock promoter, and analyzed the activation of the epitope tagged construct in the absence and presence of 50 nM furin inhibitor (Figure 3). Because cells take up injected plasmid randomly in the early embryo, we see the mosaic expression of the construct throughout the embryo in these assays. Embryos stained with anti-HA (red) and anti-GFP (green) reveal proteolytic processing of EMMA–Mmp11a by the accumulation of GFP in the absence of HA (i.e., a green signal, as opposed to an orange/yellow signal). This is more clearly illustrated in images in which the red channel has been subtracted from the green channel and the resulting data is presented as a heat map. In myocytes expressing EMMA–Mmp11a in the absence of furin inhibition, we detect activation of the construct as it traverses the secretory pathway and is released from expressing cells (Figure 3A,A′). Similarly, in epithelial cells, the loss of HA signal indicating removal of the propeptide is clearly occurring as the protein is secreted (Figure 3B,B′). Interestingly, and somewhat unexpectedly, we also see evidence of activation in the nuclei of many (but not all) cells expressing this construct (Figure 3A,B). The activation of EMMA–Mmp11a associated with the secretory pathway is markedly reduced by the furin inhibitor (Figure 3C,D). However, this does not appear to affect activation in the nuclei (Figure 3C). Immunoblots of homogenates made from embryos expressing EMMA–Mmp11a probed with anti-GFP exhibit immunoreactive bands at the expected molecular weights of both the pro- and active forms of the construct 6 h post heat shock (hpHS) (Figure 3E). However, when embryos are incubated in furin inhibitor immediately following heat shock, activation of EMMA–Mmp11a is reduced at 3 hpHS and not detectable at 6 hpHS (Figure 3E), suggesting that furin-mediated cleavage of the construct is a major contributor to the production of the activated form of the reporter.

### 3.3. Mmp11 and Timp4 Proteins Exhibit Dynamic Patterns of Accumulation in the MTJ and Muscle Cells

Using affinity purified polyclonal antibodies specific for zebrafish Mmp11a, Mmp11b, and a third polyclonal that was raised against zebrafish Timp4a, we examined the normal distribution of these antigens in embryos from 24 to 96 hpf (Figure 4). Although the antibody against Timp4a was raised against the less-conserved carboxyl domain of this protein, we cannot be certain it does not cross-react with Timp4b. So, while these data likely represent the distribution of Timp4a, we conservatively designate these simply as representative of ‘Timp4’.

Timp4 immunoreactivity is strongly and consistently detectable in the MTJs in all stages examined, and is dynamically expressed in the ectodermal epithelia of the head, fin folds, otic vesicle, the posterior notochord and muscle attachments in the jaws and pectoral fins in later stages (Figure 4A–E). By 24 hpf Mmp11a is easily detectable at the MTJs (Figure 4F, arrowhead, inset), but is subsequently detected primarily within muscle cells in the trunk myotomes and musculature of the jaw, as well as in the hatching gland, and extracellularly in the otic vesicle, fin folds, and pectoral fin mesoderm (Figure 4F–J, Figure 4J inset). With respect to its accumulation in the MTJ, Mmp11b exhibits a pattern reciprocal to that of Mmp11a, with Mmp11b absent from the MTJs at 24 hpf (Figure 4K inset) but accumulating there in later stages (Figure 4K–O, Figure 4O inset). Mmp11b also accumulates extracellularly in the otic vesicle, lens, heart and jaws in various stages. At the MTJ, the transition from Mmp11a to Mmp11b appears to occur at about 28 hpf (Figure 5). Thus, it appears that one or both paralogues of Mmp11 are present, along with Timp4 protein, at the MTJ continuously throughout this period of development.

We are able to detect both paralogues of Mmp11 extracellularly at the MTJ at 28 hpf (Figure 5), and at the maximum resolution possible with our instruments, it appears that Timp4 and the two Mmp11s are not homogeneously distributed within the ECM separating maturing myotomes. Both Mmp11 paralogues concentrate in the periphery of the MTJ, in two parallel bands closely associated with the basement membrane (Figure 5A,B), essentially identical to the pattern of laminin immunoreactivity (Figure 5D). In contrast, Timp4 immunoreactivity is present in a single strong band associated with the middle of the MTJ (Figure 5C).

The striated pattern of Mmp11a immunoreactivity we observed within myofibrils at later stages is highly suggestive of sarcomere localization. So, we used an antibody against the Z-disc protein α-actinin in these assays both as a counterstain to highlight the local cellular architecture, and to help characterize where within the muscle cells Mmp11a is accumulating. At 28 hpf the Mmp11a immunoreactivity within muscle cells appears to co-localize with α-actinin immunoreactivity (Figure 5A), and this localization at Z-discs becomes more distinct in later stages (Figure 6).

### 3.4. Mmp11 and Timp4 Proteins Interact in Specific Ways

Because of their close association at the MTJ during a period of rapid matrix remodeling, and the absence of any published data regarding the regulation Mmp11 by Timp4, we examined the potential protein-protein interactions of the catalytic and hemopexin-like domains of both Mmp11 paralogues with the N-and C-terminal domains of both Timp4 paralogues using yeast two-hybrid assays (Table 1). All assays were conducted such that each potentially interacting domain was tested as both bait and prey and using both low and high stringency selection, and only interactions that were either detected in both bait/prey configurations and/or at high stringency were considered. Despite being described as “tissue inhibitors of MMPs” the interaction of TIMPs with MMPs is not exclusively inhibitory [11,40]. The binding of the wedge-shaped N-terminal domain of a TIMP into the catalytic cleft of an MMP inhibits the proteolytic activity of the enzyme. However, the C-terminal domain of TIMPs can bind the hemopexin-like domains of MMPs, and facilitate the formation of multi-molecular complexes with substrates or activators of MMPs [67]. Thus, in addition to inhibiting MMP activity, TIMPs can also modulate MMP activity by facilitating proteolytic MMP activation and/or mediating interactions with substrates. Consistent with this modulatory role, the C-terminal domain of Timp4b interacts favorably with the hemopexin domains of both Mmp11 paralogues. The divergent C-terminal domain of Timp4a interacts with the hemopexin domain of Mmp11b, but not Mmp11a, and only in one bait-prey configuration. With respect to canonical inhibitory binding of the N-terminus of Timp4 with the catalytic domains of Mmp11, we detect robust interactions between the N-terminal domains of both Timp4 paralogues with the catalytic domain of Mmp11b. However, neither Timp4 N-terminal domain interacts detectably with the catalytic domain of Mmp11a. We therefore conclude that Timp4 likely acts as an effective inhibitor of Mmp11b, but that Mmp11a is not likely inhibited by Timp4 in vivo. Furthermore, Timp4b may function to modulate the activities of both Mmp11 paralogues; mediating their assembly into multi-molecular complexes by binding their hemopexin-like domains.

## 4. Discussion

In addition to innumerable studies linking the expression of Stromelysin-3 to the invasive and/or anti-apoptotic characteristics of tumor cells (reviewed in [14]), orthologs of Mmp11 have emerged as important effectors of matrix remodeling during amphibian metamorphosis, in which context it functions in the remodeling of the basement membrane lining the intestinal tract [15,16,17,18,19]. Like the basement membrane of the metamorphic amphibian intestine, the ECM separating embryonic somites undergoes a dramatic structural and biochemical remodeling during the second day of embryonic development in the zebrafish, transitioning from a rectangular fibronectin-dominated somite boundary into a chevron-shaped laminin/collagen-dominated myotendinous junction (MTJ) [5,6,7]. Mmp11 becomes concentrated in the MTJ during this time period, where it is both necessary and sufficient to mediate the removal of fibronectin [12]. Interestingly, the concentration of Mmp11 at the MTJ is sensitive to the assembly of laminin-rich basement membranes in this context, suggesting that laminin-signaling acts as a checkpoint for this developmental ECM remodeling event [12]. We can now add to our understanding of this process the fact that Timp4 is also abundant at these boundaries prior, during and after this transition, and that it is Mmp11a that is there initially, but Mmp11b that is present later. Furthermore, since neither Timp4 paralogue appears to act as an inhibitor of Mmp11a, it seems likely that Timp4 present at the MTJ early is acting as a mediator/modulator of early Mmp11 activity—possibly activating and/or facilitating the interaction of Mmp11a with its substrate(s)—whereas the activity of Mmp11b present later in development may be inhibited and kept in check by Timp4 (Figure 7).

The hypothesis that Mmp11 directly degrades fibronectin in the maturing MTJ is an attractively parsimonious explanation for the data reported by Jenkins et al. [12]. However, mammalian MMP11 protein shows little proteolytic activity against fibronectin in vitro [39,51,52,68]. It may be that the MMP11 binding sites and/or scissile bonds in fibronectin are only exposed when the fibrils are under mechanical load. Alternatively, cofactors that were not present in the in vitro assays may be necessary for MMP11 to degrade fibronectin. Although it does not support the activation of mammalian MMP2, TIMP4 binds the hemopexin-like domain of MMP2, and may function as a cofactor in its interactions with substrates [69,70]. Based on our yeast two-hybrid data, we speculate that Timp4 may be such a cofactor for Mmp11, possibly mediating an interaction between the hemopexin-like domain of Mmp11a and fibronectin and thereby facilitating the removal of fibronectin from the nascent MTJ.

In the absence of purifying selection, many paralogous genes resulting from the teleost genome duplication event have been lost [71,72]. However (and unlike genes encoding other MMPs and TIMPs, such as *mmp2* and *timp3* which are present as single copies), both copies of *mmp11* and both copies of *timp4* have been retained in the zebrafish genome. Moreover, these paralogues have accumulated mutations that likely alter their biochemistry significantly, suggesting selective pressures for divergent functions [72]. This is further supported by the distinctly different patterns of protein accumulation observed for Mmp11a versus Mmp11b.

That both Mmp11 paralogues are activated by furin is consistent with predictions based on the conserved paired basic amino acid motif recognized by furin proprotein convertases [38,39,73] present in their propeptides. However, this assumption has not previously been tested in vivo. Furthermore, the intracellular localization of both the endogenous and epitope tagged Mmp11a proteins and intracellular activation of EMMA–Mmp11a in the presence of a strong furin inhibitor suggests that furin-independent mechanisms may also play roles in the regulation of Mmp11a at least, and perhaps the regulation of other predicted furin targets. We cannot rule out the possibility that the overexpression of EMMA–Mmp11a is simply overwhelming the cellular secretory mechanism, but evasion of the secretory pathway and subsequent cytoplasmic or nuclear localization has been described for a variety of MMPs [47,49,59,61,74,75,76,77,78], including MMP11 [60,79]. And unless the activated EMMA–Mmp11a we detect in nuclei reached that location through an unknown mechanism, it would not have come into contact with furin and must therefore have been activated by some other mechanism. Furthermore, the furin recognition motif may also function as a nuclear localization signal in this context [80]. The intracellular roles of MMPs remain poorly understood, apart from increasing evidence that these proteases become activated under pathological conditions during which they participate in a variety of mechanisms resulting in cellular damage and death [45,46,47,48,77,81]. However, it seems unlikely that highly conserved mechanisms that concentrate these potentially dangerous proteases in structures like sarcomeres do not serve some physiologically advantageous processes of which we remain ignorant [49]. We suggest that Mmp11a may be playing important, unrecognized physiological roles within the sarcomeres and/or nuclei of cells under normal conditions, and that furin-independent mechanism(s) regulate its activation in these contexts.

While TIMPs 1 and 2 are relatively well-studied, TIMPs 3 and 4 remain poorly characterized, and frustratingly little is known about the functions of any of the TIMPs in vivo [41]. One recent in vitro study suggested that recombinant TIMP4 forms complexes with Stromelysin-3 from canine carcinoma cells and reduces its activity in gel zymography assays [82], but it is unclear why the relative mobility of the Stromelysin-3 in these assays is not altered when ostensibly in complex with TIMP4, drawing these results into question. Analyses comparing MMP activity in vivo to activity in homogenates of same-stage embryos in vitro elegantly demonstrate the presence of abundant but heterogeneously distributed MMP inhibitors in zebrafish embryos [22], but this does not shed any light on which endogenous inhibitors are interacting with which MMPs. Here, we show that, in addition to being present at the same time and place in the embryo, both Timp4 proteins interact with both Mmp11 proteins, albeit in distinctly different ways, suggesting that Timp4 is an important regulator of Mmp11-mediated matrix remodeling at the MTJ in vivo. Intriguingly, neither paralogue of Timp4 appears to interact with Mmp11a via the canonical inhibitory mechanism (whereby the N-terminus of the TIMP binds the catalytic domain of the MMP). This suggests that the Mmp11a-Timp4 interactions occurring during the early phase of MTJ remodeling may facilitate the formation of protein complexes that allow Mmp11 to act on specific substrates (such as fibronectin). We detect canonical inhibitory interactions between the catalytic domain of Mmp11b and the N-terminal domains of both Timp4 paralogues, and so we would predict that, after 28 hpf as Mmp11a is replaced by Mmp11b at the MTJ, proteolytic activity is both targeted and constrained by Timp4, such that the system functions to maintain the integrity of the mature boundary. However, the differences in sequence and predicted structures of the Mmp11 paralogues, as well as the highly charged insertion in the N-terminal domain and reduced C-terminal domain of Timp4a, all suggest that these proteins have functionally diverged. So, this model remains speculative and in need of further experimental support.

MMP-mediated matrix remodeling underpins the emergence of tissue architecture during embryonic development, the regeneration of functional tissues during wound healing, and its mis-regulation is fundamental to an enormous array of pathologies [10,11,83,84,85,86]. Most of what is known about most MMPs derives from in vitro studies with a focus on pathological mechanisms. The zebrafish embryo provides a powerful model system in which to investigate the developmental and physiologically relevant activities of this and other MMPs in vivo, and the molecular mechanisms regulating them [9,21,22,25,27,31]. Here, we have utilized this system to expand our understanding of the roles of Mmp11 and its regulation by Timp4 in the context of MTJ remodeling and suggest that there is a great deal more to be learned using this approach.

## Figures and Tables

**Figure 1 jdb-07-00022-f001:**
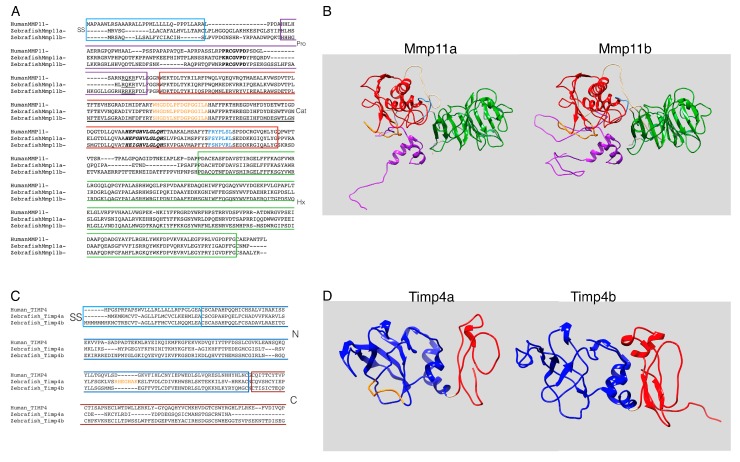
Sequences and predicted structures of zebrafish metalloproteinase 11 (Mmp11) and tissue inhibitors of metalloproteinase-4 (Timp4) paralogues exhibit both conserved and divergent features. (**A**) Inferred amino acid sequences of zebrafish Mmp11a and Mmp11b paralogues aligned with human MMP11 for reference. Secretory signals are boxed in light blue, propeptide in purple (with cysteine switch motif highlighted in bold and furin recognition sequence underlined), catalytic domain in red (with S-loop in orange, zinc binding motif in bold italic, and specificity loop in light blue), and the carboxyl hemopexin domain is boxed in green. (**B**) Structural homology models of zebrafish Mmp11a and Mmp11b rendered as ribbon diagrams and colored as in the sequence alignment. (**C**) Inferred amino acid sequences of zebrafish Timp4a and Timp4b paralogues aligned with human TIMP4 for reference. Secretory signals are boxed in light blue, the N-terminal domain in blue (with the seven-residue charged insertion of Timp4a highlighted in orange), and the C-terminal domain in red. (**D**) Structural homology models of zebrafish Timp4a and Timp4b rendered as ribbon diagrams and colored as in the sequence alignment.

**Figure 2 jdb-07-00022-f002:**
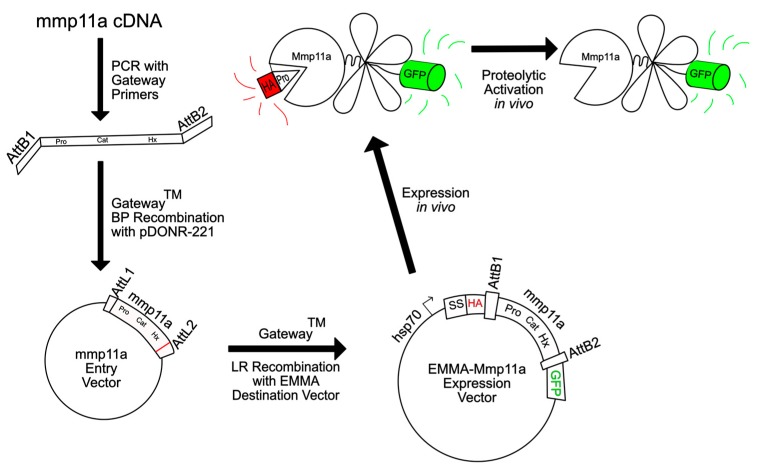
Schematic illustration of the construction of EMMA–Mmp11a. The coding sequence of zebrafish *mmp11a* 3′ to the secretory signal was amplified using primers with Gateway recombination sites, recombined into a donor vector, and from there into the EMMA destination vector. This allows the expression of a doubly epitope-tagged protein, whereby loss of co-localization of the HA and GFP epitopes signifies proteolytic activation of the protease by endogenous mechanisms [25].

**Figure 3 jdb-07-00022-f003:**
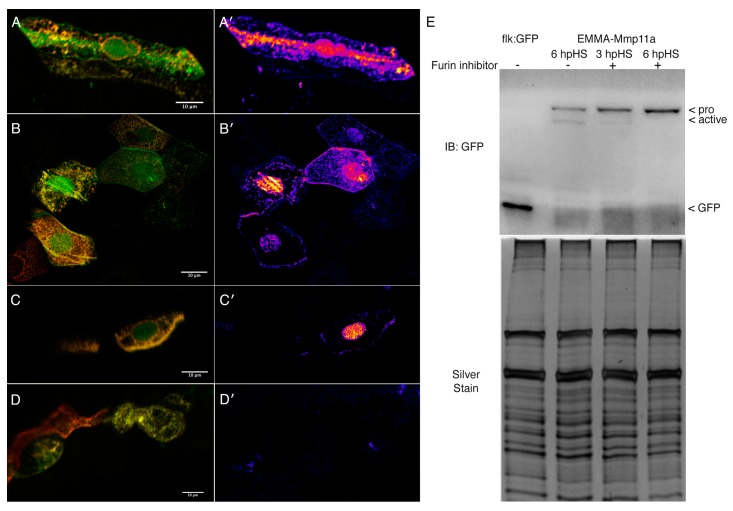
Mmp11a is activated by furin except in the nucleus. (**A**) Myocytes expressing EMMA–Mmp11a remove the hemagglutinin (HA)-tagged propeptide (red channel), leaving only the carboxyl terminal GFP tag (green channel) as the construct traverses the secretory pathway, and also in the nucleus. This is more clearly visible in the false color heat map showing the strength of the GFP signal relative to the HA signal (**A′**). (**B**) Epithelial cells expressing EMMA–Mmp11a also remove the propeptide as the construct is secreted and released extracellularly, as well as in the nucleus. (**B′**) Heat maps of the relative abundance of GFP vs. HA clearly illustrate the activation of the EMMA–Mmp11a construct as it is secreted, as well as within the nuclei. (**C**) Myocytes expressing EMMA–Mmp11a in the presence of furin inhibitor only remove the propeptide in the nuclei. (**C′**) Heat maps of myocytes expressing the construct in the presence of the furin inhibitor show relatively unimpaired proteolytic removal of the propeptide in the nuclei, but dramatically reduced activation in the secretory pathway and extracellularly. (**D**) Epithelial cells expressing the construct in the presence of furin inhibitor also show dramatically reduced activation. (**D′**) Heat maps of epithelial cells in the presence of furin inhibitor, showing negligable activation of the contstruct. (**E**) Immunoblots of embryo homogenates expressing either GFP alone (flk:GFP), or EMMA–Mmp11a, in the presence or absence of furin inhibitor, probed with anti-GFP. Activation of the construct is clearly detectable in embryos 6 h post heat shock (hpHS) in the absence of inhibitor. In the presence of inhibitor, some activation is detectable 3 hpHS, but by 6 hpHS activation, appears completely abolished. Silver staining of a replicate gel shows comparable protein loads. Scale bars are 10 µm.

**Figure 4 jdb-07-00022-f004:**
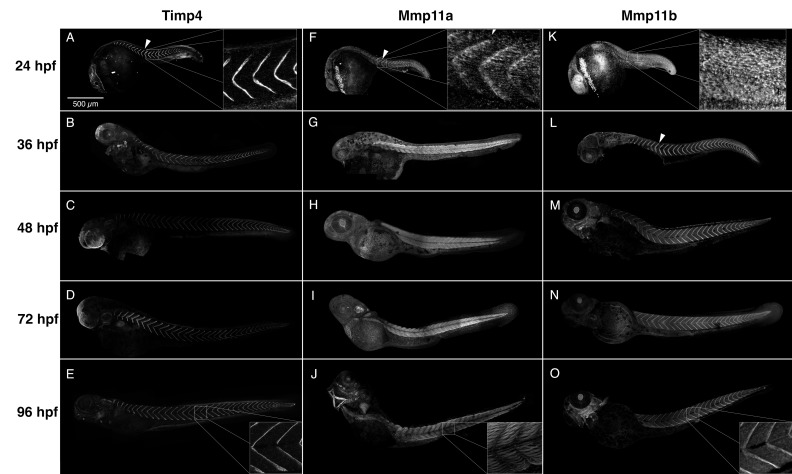
Timp4 and Mmp11 paralogues accumulate dynamically during development and co-localize at the myotendinous junctions (MTJs). Composite confocal projections of 24, 36, 48, 72 and 96 hpf embryos labeled with antibodies against Timp4, Mmp11a, or Mmp11b. (**A**–**E**) Timp4 immunoreactivity is abundant at the MTJ at all stages examined (arrowhead), and accumulates dynamically in ectodermal epithelia of the head, fin folds, otic vesicle, the posterior notochord and muscle attachments in the jaws and pectoral fins in later stages. (**F**–**J**) Mmp11a is present at the MTJs in 24 hpf embryos (**F**) (arrowhead) but becomes localized within muscle cells in later stages (**G**,**H**). (**K**–**O**) Mmp11b becomes concentrated in MTJs at later stages (**L**–**O**) (arrowhead). Insets show higher magnification views of MTJs dorsal to the yolk extension in 24 and 96 hpf embryos for comparison. Scale bar = 500 µm.

**Figure 5 jdb-07-00022-f005:**
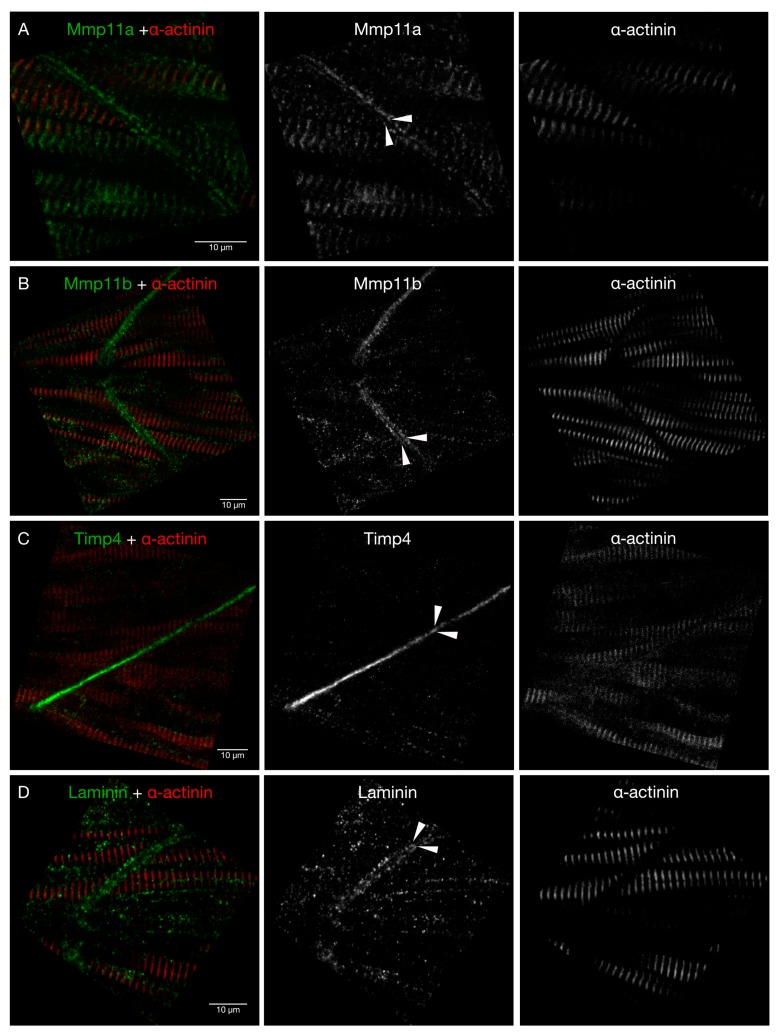
Timp4 and both Mmp11 paralogues are present in the MTJ at 28 hpf. High-resolution confocal micrographs of MTJs in the trunk skeletal musculature dorsal to the yolk extension of 28 hpf embryos stained with antibodies against (**A**) Mmp11a, (**B**) Mmp11b, (**C**) Timp4, and (**D**) Laminin (green) as well as anti-α-actinin (red) reveal that both Mmp11 paralogues are detectable in MTJs at this stage, and that they accumulate in the periphery of the MTJ (arrowheads), whereas Timp4 accumulates in the core of the MTJ. Scale bars are 10 µm. Anterior to the left in all images.

**Figure 6 jdb-07-00022-f006:**
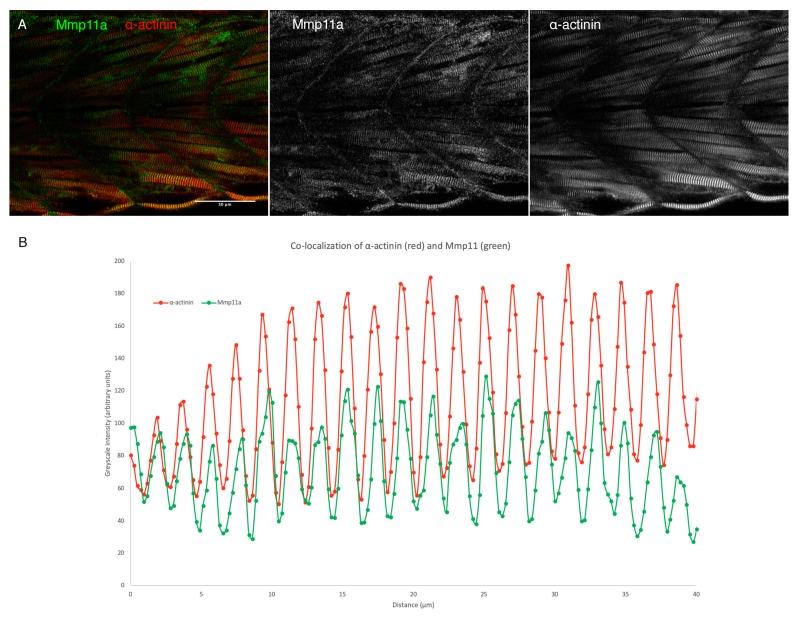
Mmp11a localizes to the Z-discs of skeletal muscle sarcomeres. (**A**) Confocal micrographs through the trunk musculature of a 48 hpf embryo double labeled with anti-Mmp11a (green) and anti-α-actinin (red) reveal co-localization of the antigens. (**B**) Quantification of signal intensity in both channels reveals precise co-localization of Mmp11a with α-actinin. Scale bar 50 µm. Anterior to the left.

**Figure 7 jdb-07-00022-f007:**
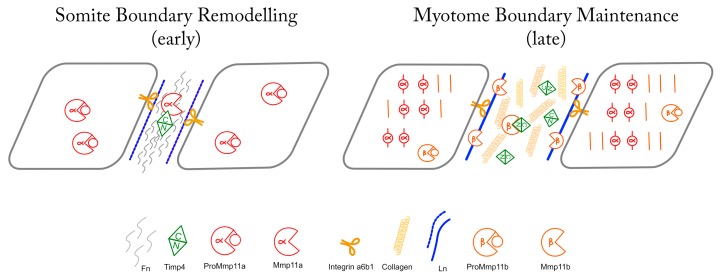
Schematic representation of hypothesized activities and interactions between Mmp11 and Timp4 paralogues at early and late MTJs. The hemopexin-like domain of Mmp11a interacts with the C-terminal domain of Timp4b in the early MTJ, facilitating the degradation of fibronectin. In the mature MTJ, Mmp11b is inhibited by Timp4a, constraining its activity in the maintenance of the ECM, while Mmp11a localizes to the Z-discs of the sarcomeres within the myocytes.

**Table 1 jdb-07-00022-t001:** Interactions between domains of Mmp11 and Timp4 detected by yeast two-hybrid assay.

	Timp4a-N	Timp4a-C	Timp4b-N	Timp4b-C
**Mmp11a-Cat**	--	-	--	--
**Mmp11a-Hx**	-	--	--	++
**Mmp11b-Cat**	+	--	++	--
**Mmp11b-Hx**	-	-	-	+

Key: Domains that did not interact or interacted only weakly and in only one bait–prey combination are indicated with ‘--’ or ‘-’, respectively. Positive interactions are indicated with ‘+’ or ‘++’ for interactions that were robust and detected in both bait–prey combinations. Canonically, inhibitory interactions between the N-terminal domain of the Timp4 protein and the catalytic domain of the Mmp11 protein are shaded red, while putative modulatory interactions between the C-terminal domain of the Timp4 protein and the hemopexin-like (Hx) domain of the Mmp11 protein are shaded green.

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
