# Peer review of "Paralogues of Mmp11 and Timp4 Interact during the Development of the Myotendinous Junction in the Zebrafish Embryo"

_jdb, 2019, doi:10.3390/jdb7040022_

Round 1

Reviewer 1 Report

Crawford jdb-608248 – review

In this manuscript members of the Crawford lab undertake a precise analysis of the interactions and functions of paralogues of MMP11 and TIMP4 in zebrafish, focusing on myotendinous junction formation. They employ online tools to compare the structures and domains of these paralogues to highlight differences. They then examine the importance of these differences through a number of assays. The differences in the furin recognition site in the MMP11 paralogues was shown to impact their activation – demonstrated through their use of an EMMA assay. Confocal image constructs following immunolocalization were used to demonstrate the shifting localization of TIMP4, MMP11a and MMP11b within the developing fish embryo – with focus being put on the somites, and related to laminin deposition. The study concluding with a demonstration that the MMP11 and TIMP4 paralogues have very specific binding relationships, as demonstrated using bait/prey combinations that demonstrated that these TIMPs could not bind to the catalytic domain of MMP11a, both TIMP4a and b N-terminal domains could bind to the catalytic domain of MMP11b, while the C-terminal domain of TIMP4c could bind to the hemopexin domains of both MMP paralogues.
Together their data demonstrates clearly the complexity and timeliness of MMP and TIMP functions. While the remodelling of the ECM is a clear function, their data demonstrates that these molecules very likely signal and interact to regulate many developmental processes – a finding that is of broad interest.

I find no major flaws in the experimental design, data, nor its interpretation. However, there are items that can be clarified.

1) The source of the laminin and α-actinin antibodies is not mentioned in the methods.
2) The EMMA methodology could be better explained (a figure?).
3) Microinjections involve 3-5nl of plasmid – an amount of plasmid is needed (not just volume).
4) The MMP11 paralogues are most often identified as a and b, but at times they are labelled α and β. Naming needs to be consistent.
5) in the first paragraph of the results a “specificity” domain is mentioned, that is not defined. Is this the hemopexin domain?
6) Furin inhibitor was used to investigate MMP11 activation. However, the use of a furin inhibitor would have many effects on many proteins, potentially impacting many signalling pathways and development. Can the authors comment on the level of furin inhibitor used? Has this level been used in other fish studies? Did the embryos display developmental defects of this time course?
7) Figures 3 and 4 proport to demonstrate the localization of proteins in the MTJ. However the magnification in 3 is not sufficient to clearly seem the MTJ (perhaps a magnified insert), while with figure 4 there is no sense of orientation (perhaps a less magnified diagram of the area).
8) The authors may need to clarify the use of the term “in” when describing data from figures 3 and 4. They use “within” the muscle cells – and indeed they mean in the cell, but then they state “in” the otic vesicle, lens, heart etc. Do they mean in the cells? Or are these secreted proteins in the ECM within these tissues?

Author Response

We would like to thank the Reviewers for their time and constructive feedback on our submission, and to apologize for the delay in submitting our revision addressing these issues.  Most of this was due to delays in getting reviewer feedback regarding our submission to biomedicines, which is reference 49 in the present manuscript.  That paper has now been accepted for publication, and we will complete missing bibliographic information in the References section as soon as that information is available (which we expect will be only a few days).  

We believe we have have addressed all of the issues raised by all Reviewers, with specific changes outlined below.

Reviewer 1

Missing antibody information has been added. We have added a schematic (Fig 2) illustrating the process of generating the EMMA-Mmp11a construct and the tagged proteins. Plasmid concentrations and final injected amounts have been added. Corrected. The "S-loop" and Specificity loop are sections of the catalytic domain of MMPs best described in Maskos, K., & Bode, W. (2003). Structural basis of matrix metalloproteinases and tissue inhibitors of metalloproteinases. Molecular Biotechnology, 25(3), 241–266. http://doi.org/10.1385/MB:25:3:241; we have added this reference to clarify this discussion of the structural features of Mmp11 (new reference 65). Yes, we expect the inhibition of furin activity would have many effects on embryonic development, however, we did not observe any developmental defects during the few hours embryos were exposed in these experiments.  Because this is quite an expensive reagent, we did not pursue experiments over longer time courses which would have required refreshing the reagent by replacing the solution in which the embryos were incubating. We would certainly predict that it would perturb embryonic development given enough time.  However, as the reviewer correctly points out, there are so many processes that would be disrupted in such an experiment, it would be difficult to interpret such results. While we are unaware of other studies using this inhibitor on zebrafish embryos, it has been used extensively in vitro, and we used the inhibitor at 50 nM because it has been used intravenously in mice at comparable concentrations. We have added magnified insets to the 24 and 96 hpf panels of the new Figure 4 (which was previously Figure 3), and we have re-done Figure 5 (which was Figure 4) such that all panels are in the same orientation (anterior to the left) for easier comparison.  All of the images in Figure 5 are of the skeletal musculature dorsal to the yolk extension, and this is now clarified in the text. The Reviewer is entirely correct that this is can be ambiguous; in some contexts it is not clear wether the antigen is extracellular or intracellular. But we have edited the text throughout to try to be more explicit where possible.

Reviewer 2 Report

The work by Matchett and colleagues provides new information about the myotendinous junction at the interface between somites development and the role of Mmp11 and Timp4 in a zebrafish model system. Of special interest is the provisional extracellular matrix (ECM) that is rich in fibronectin, which is replaced by an ECM that contains laminin and collagens. Authors demonstrate that Mmp11a and Mmp11b localize to the MTJ during development but at different times.

Minor

Page 3 line 139: capitalize ‘Construction’

Page 4, line 156: please delete one closing parentheses; currently, there are three, and only two are needed

Page 4, line 168: please capitalize PAGE

Page 5, line 203: change period to a colon.  Currently, the statement that starts with  “The crystal structure….” is not a complete sentence. I believe that you want to present a list of structures that are used for modeling. Alternatively, you could add the phrase, “…were used” at the end to correct sentence structure.

Reference 49 is not yet published. Please update and only cite published works.

Author Response

Response to reviewers

We would like to thank the Reviewers for their time and constructive feedback on our submission, and to apologize for the delay in submitting our revision addressing these issues.  Most of this was due to delays in getting reviewer feedback regarding our submission to biomedicines, which is reference 49 in the present manuscript.  That paper has now been accepted for publication, and we will complete missing bibliographic information in the References section as soon as that information is available (which we expect will be only a few days).  

We believe we have have addressed all of the issues raised by all Reviewers, with specific changes outlined below.

Reviewer 2

All issues identified by this Reviewer have been corrected in the text. Reference 49 has now been accepted for publication; we will update the reference with complete bibliographic information as soon as it is available (which should be in the next few days).